# Evaluation of the Radiation Shielding Properties of a Tellurite Glass System Modified with Sodium Oxide

**DOI:** 10.3390/ma15093172

**Published:** 2022-04-27

**Authors:** Khalid I. Hussein, Mohammed S. Alqahtani, Arwa A. Meshawi, Khloud J. Alzahrani, Heba Y. Zahran, Ali M. Alshehri, Ibrahim S. Yahia, Manuela Reben, El Sayed Yousef

**Affiliations:** 1Department of Radiological Sciences, College of Applied Medical Sciences, King Khalid University, Abha 61421, Saudi Arabia; mosalqhtani@kku.edu.sa (M.S.A.); arwa155_@hotmail.com (A.A.M.); 437808203@kku.edu.sa (K.J.A.); 2Department of Medical Physics and Instrumentation, National Cancer Institute, University of Gezira, Wad Medani 2667, Sudan; 3BioImaging Unit, Space Research Centre, Department of Physics and Astronomy, University of Leicester, Leicester LE1 7RH, UK; 4Physics Department, Faculty of Science, King Khalid University, Abha 61413, Saudi Arabia; dr_hyzahran@yahoo.com (H.Y.Z.); amshehri@kku.edu.sa (A.M.A.); dr_isyahia@yahoo.com (I.S.Y.); omn_yousef2000@yahoo.com (E.S.Y.); 5Research Center for Advanced Materials Science (RCAMS), King Khalid University, Abha 61413, Saudi Arabia; 6Nanoscience Laboratory for Environmental and Bio-Medical Applications (NLEBA), Semiconductor Lab., Metallurgical Lab. 2, Physics Department, Faculty of Education, Ain Shams University, Roxy, Cairo 11757, Egypt; 7Faculty of Materials Science and Ceramics, AGH—University of Science and Technology, Al. Mickiewicza 30, 30-059 Krakow, Poland; manuelar@agh.edu.pl

**Keywords:** phospho-tellurite glass, molar volume, half-value layer, mass attenuation coefficient, optical properties

## Abstract

In this study, the X-ray and gamma attenuation characteristics and optical properties of a synthesized tellurite–phosphate–sodium oxide glass system with a composition of (85 − x)TeO_2_–10P_2_O_5_–xNa_2_O mol% (where x = 15, 20, and 25) were evaluated. The glass systems we re fabricated by our research group using quenching melt fabrication. The shielding parameters of as-synthesized systems, such as the mass attenuation coefficient (MAC), linear attenuation coefficient (LAC), effective atomic number (Z_eff_), half-value layer (HVL), tenth value layer (TVL), mean free path (MFP), and effective electron density (N_eff_) in a wide energy range between 15 keV and 15 MeV, were estimated using well-known PHY-X/PSD software and recently developed MIKE software. Herein, the optical parameters of prepared glasses, such as molar volume (V_M_), oxygen molar volume (V_O_), oxygen packing density (OPD), molar polarizability ($\alpha_{m}$), molar refractivity (R_m_), reflection loss (R_L_), and metallization (M), were estimated using MIKE software. Furthermore, the shielding performance of the prepared glasses was compared with that of commonly used standard glass shielding materials. The results show that the incorporation of sodium oxide into the matrix TeO_2_/P_2_O_5_ with an optimum concentration can yield a glass system with good shielding performance as well as good optical and physical properties, especially at low photon energy.

## 1. Introduction

The possible threats to live biological cells due to the harmful effects of ionizing radiation should be properly addressed. Radiation shielding has grown increasingly crucial in recent years as we have learned more about the biological effects of ionizing radiation. Ionizing radiation, such as gamma rays and X-rays, has been used for a variety of applications, including diagnostic imaging, radiotherapy, nuclear medicine, nuclear reactors, industrial operations, and food safety. The goal of radiation shielding is to eliminate (or reduce) human and environmental exposure to radiation. The considerably more prevalent gamma shielding material in ionizing radiation facilities is lead (Pb). Lead has a number of disadvantages, including toxicity, a lack of transparency, and poor substance properties. Transparent radiation shielding materials, as opposed to opaque radiation shielding materials, serve an essential role in nuclear engineering research because they provide substantial radiation protection while still being visible [1]. Recently, glasses have been considered as an alternative radiation shielding material due to properties such as their transparency, ease of manufacturing, nontoxicity, and high density [1,2,3,4,5,6,7,8]. Glass must be uniform in density and composition when used as a radiation shielding material. The addition of specific types of oxides to a glass matrix may increase glass formation and the structural, optical, thermal, and shielding properties. Phosphate-based glasses have aroused the interest of researchers due to their distinctive properties, such as their good thermal expansion coefficients, low melting temperatures, transparency, good optical properties, and chemical stability, which allow them to be easily prepared as hosts for metal doping in various applications [9,10].

Tellurium oxide (TeO_2_) is also an essential oxide for glass synthesis, which requires rapid quenching to produce glass structures. Because of their high dielectric constants, high glass-forming capacity, low melting point, chemical stability, and highly nonlinear optical properties, tellurite glass systems have received a lot of interest in recent decades [11,12,13,14]. Tellurite-based glasses are used in a variety of applications, including thermal electrical equipment, optical amplifiers, gas sensors, and radiation shielding glasses [15,16,17,18]. Several studies and investigations have recently been conducted to better understand the chemical, optical, physical, and structural properties of TeO_2_-based glasses modified with alkaline, rare-earth oxides, or transition-metal oxides [19,20,21,22,23], which have been improved in terms of their high refractive indices and high thermal expansion stability.

Yousef [23] studied the incorporation of sodium oxide into a TeO_2_/P_2_O_5_ matrix. His findings revealed a modified glass with superior thermal stability, transparency, glass transition temperature, and crystallization stability. In addition to these features, various investigations have found that TeO_2_-based glasses are a good choice for ionizing radiation shielding. To put it another way, TeO_2_-based glasses have outstanding radiation shielding properties due to their high molecular weight [11,12]. It is possible to significantly improve the optical properties of glasses by adding modifiers such as alkali ions.

The purpose of this study is to examine the gamma shielding effectiveness, optical properties, and thermal stability of a new composition glass system developed by our research group [23] with the composition of (85 − x)TeO_2_–10P_2_O_5_–xNa_2_O mol% (where x = 15, 20, and 25). The radiation and optical parameters were calculated using PHY-X/PSD software [24] and recently developed MIKE software [25]. The prepared glasses were compared with some commercial standard radiation shielding materials [26].

## 2. Materials and Methods

Our research group [25] utilized quenching melt fabrication to synthesize glasses with the composition of (85 − x)TeO_2_–10P_2_O_5_–xNa_2_O mol% (where x = 15, 20, and 25) [23]. The density values of the prepared glasses were taken from [23]. The glass systems were synthesized using the melt quench technique. Specific weights of raw metal oxides (TeO_2_, P_2_O_5_, and Na_2_O of purity ≥ 99%) were mixed and placed in a platinum crucible and heated in a melting furnace to temperatures ranging from 850 to 900 °C for 30 min. The furnace was switched off, and the sample was allowed to cool to room temperature. Thermal analysis and thermal stability were also carried out according to [23]. Using the cadmium lamp spectrum, a prism spectrometer (a V-block Pulfrich refractometer PR2, Carl Zeiss, Jena, Germany) was used to measure the refractive index of the studied samples at a wavelength of 479.98 nm. A UV–VIS–NIR spectrophotometer was used to measure the optical absorption spectra at wavelengths ranging from 200 to 2500 nm (JASCO V-570, Tokyo, Japan). The radiation parameters were calculated using PHY-X/PSD and MIKE software [24,25]. The shielding parameters were compared with commonly used standard radiation shielding materials such as RS-253-G18, RS 360, and RS 520 [26]. RS-360 and RS-520 are considered the more effective radiation shielding glasses because of their high PbO contents of 45% and 71%, respectively.

### 2.1. Optical Properties

The molar volume is the volume occupied by one mole of any material at a given temperature and pressure. The following formula is used to compute the molar volume (V_M_) of a material for a given composition and density [27]:(1)$$V_{M}=\frac{M_{w}}{\rho }$$
where M_w_ is defined as the total molar weight of the sample, and ρ is the density of the sample.

The parameter V_O_, which can be derived using the following equation [27], measures the volume of glass in 1 mole of oxygen:(2)$$V_{o}={\;V}_{M}\left(\frac{1}{\sum x_{i}n_{i}}\right)$$
where $n_{i}$ is the number of oxygen atoms in each oxide. The oxygen packing density (OPD) of any glass material can be computed using the following equation [27], which represents the density and V_M_ characteristics according to the chemical bond approach:(3)$$\mathrm{OPD}=1000\sum x_{i}n_{i}\left(\frac{1}{V_{M}}\right)$$

The molar refractivity (R_m_) can be used to determine the overall polarizability of a mole of a material, which is used to investigate the role of ionic packing in influencing the refractive indices of glass materials.

The following equation [27] can be used to calculate R_m_:(4)$$R_{m}=\left(\frac{n^{2}-1}{n^{2}+2}\right)V_{m}$$

The reflection loss (R_L_) can be calculated using Fresnel’s formula [28]:(5)$$R_{L}={\left[\frac{\left(n-1\right)}{\left(n+1\right)}\right]}^{2}$$

The molar polarizability of the glass ($\alpha_{m}$) is proportional to R_m_, and it can be calculated using the following formula [28]:(6)$$\alpha_{m}=\left(\frac{3}{4{\pi N}_{A}}\right)R_{m}$$
where N_A_ is Avogadro’s number.

The metallization (M), which can be used to assess whether a substance is metallic or nonmetallic, is given by the following formula [28]:(7)$$M=1-\frac{R_{m}}{V_{m}}$$

If R_m_/V_m_ < 1 (i.e., M > 0), the materials demonstrate an insulating nature, but if R_m_/V_m_ > 1 (i.e., M < 0), the materials show a metallic nature.

The dielectric constant (ε) and optical dielectric constant (ε_O_), as functions of the refractive index, can be given by the following equation [28]:ε = n^2^(8)
ε_O_ = n^2^ − 1(9)
where n is the refractive index.

### 2.2. Radiation Shielding Parameters

A gamma photon is attenuated when it passes through a specific material thickness. For a given thickness, a better shielding material will have a higher attenuation. The degree of attenuation depends on various photon interaction processes. The attenuation coefficient can be estimated using the Lambert–Beer law [29,30]:(10)$$I_{x}=I_{0}e^{-\mu x}$$
where I_0_, I_x_, µ, and x denote the intensity of incident radiation, the transmitted radiation intensity, the linear attenuation coefficient, and the absorber thickness, respectively.

The mass attenuation coefficient (MAC) and linear attenuation coefficient (LAC) can be theoretically estimated using the mass attenuation of elemental compositions of the prepared glass sample. The following equations can be used to calculate the MAC and the LAC for a given energy [31,32]:$$\mathrm{MAC}=\frac{\mu }{\rho }=\sum_{i}w_{i}{\left(\frac{\mu }{\rho }\right)}_{i}$$
(11)LAC = MAC ×ρ
where $w_{i}$ is the fraction by weight of the ith atomic element, ${\left(\frac{\mu }{\rho }\right)}_{i}$ is the mass attenuation of the ith atomic element, and ρ is the density of prepared glasses.

The half-value layer (HVL) and tenth value layer (TVL) are defined as the desired thicknesses at which the attenuated intensities are 50% and 90% of the narrow photon beam intensity, respectively. The HVL and TVL shielding characteristics are inversely proportional to the linear attenuation of the shielding material. As a result, the following equations can be used [31,32]:(12)$$\mathrm{HVL}=\frac{0.693}{\mu_{\mathrm{LAC}}}\left(\mathrm{cm}\right)$$
(13)$$\mathrm{TVL}=\frac{2.303}{\mu_{\mathrm{LAC}}}\left(\mathrm{cm}\right)$$

The mean free path (MFP) is the measure of the distance traveled between two successive gamma-ray collisions and can be calculated as follows [33]:(14)$$\mathrm{MFP}=\frac{1}{\mu_{\mathrm{LAC}}}\left(\mathrm{cm}\right)$$

The effective atomic number (Z_eff_) is the term used to describe the attenuation of gamma rays that happens as a result of partial photon interactions with matter, and it is represented by the following equation [34,35]:(15)$$Z_{\mathrm{eff}}=\frac{\sum i\;f_{i}A_{i}{\left(\frac{\mu }{\rho }\right)}_{i}}{\sum j\;f_{j}A_{j}{\left(\frac{\mu }{\rho }\right)}_{j}}$$
where $\left(f_{i},{\;A}_{i}\right):$ is the fractional abundance $\left(\sum i\;f_{i}=1\right)$ and the atomic weight, respectively.

The effective electron density ($N_{\mathrm{eff}}$) is determined by the relation [36]:(16)$$N_{\mathrm{eff}}=N_{A}\frac{{\mathrm{nZ}}_{\mathrm{eff}}}{\sum i\;n_{i}A_{i}}\left(\mathrm{electron}/g\right)$$

## 3. Results and Discussion

### 3.1. Physical and Optical Parameters

Thermal analysis and thermal stabilization of the tested glass materials were performed according to [23]. Table 1 shows the composition, density, and refractive index of the samples under investigation. The density and refractive index decreased when the concentration of Na_2_O increased from 15 to 25 mol%. The refractive index decreased from 2.128 to 2.068, and the density decreased from 4.602 to 4.149 gm.cm^−3^. These results are consistent with previous results [37] and were due to the increased Na_2_O concentration and the decreased concentration of TeO_2_. On the other hand, the inclusion of alkaline oxides in the glass network resulted in high thermal stability due to changes in the structure of the glass network as the concentration of Na_2_O increased, which resulted in a high glass transition temperature and improved crystallization stability.

The samples’ molar volume (V_M_), oxygen molar volume (V_O_), and oxygen packing density (OPD) are shown in Table 2. The V_M_ and V_O_ increased as the concentration of Na_2_O increased; however, the OPD decreased, meaning that the glass structure became tighter with fewer connections in the matrix, in contrast to the density. Both V_M_ and V_O_ were proportional to the spatial distributions of oxygen in the glass matrix, increasing from 30.923 to 31.950 cm^3^ and 13.445 to 14.521 cm^3^·mol^−1^, respectively. As the Na_2_O concentration increased from 15 to 25 mol%, the OPD value dropped from 74.37 to 68.86 mol·dm^−**3**^.

The TPN glasses were evaluated for optical absorption throughout a wavelength range of 250 to 2500 nm, as shown in Figure 1. As can be seen, the absence of a strong absorption edge in the spectra confirms the amorphous nature of the TPN samples and indicates the non-crystallization of the studied materials. Furthermore, as illustrated in Figure 1, increasing the concentration of sodium oxide (Na_2_O) increased the absorbance. TPN3 had the greatest absorbance values in the visible spectrum of light, while TPN2 had the lowest absorbance and lowest Urbach energy with the highest molar refractivity and electronic polarizability, implying that it is suitable for optical applications such as nonlinear waveguides and gain media doped with rare earth for producing laser sources and fiber optics.

The molar refractivity (R_m_), reflection loss (R_L_), molar electronic polarizability (α_m_), metallization, dielectric constant (ε), and optical dielectric constant (ε_O_) for the investigated glasses were estimated using the measured refractive index. Table 3 illustrates these values. As the refractive index decreased with the increase in Na_2_O from 15 to 25 Mol%, the molar refraction, reflection loss, and molar electronic polarizability decreased from 16.714 to 16.678, 0.130 to 0.121, and 6.633 to 6.618, respectively. Furthermore, the modifier Na_2_O at 20 mol% concentration in the glass matrix created a fraction of the distorted TeO_4_ tbp phase, which had non-oxygen bridges (NBOs) with different bond lengths and also TeO_3_ tp with two NBOs, which resulted in high values for molar refractivity and consequently high electronic polarizability values [38]. The dielectric and optical dielectric constants decreased with increases in the Na_2_O concentration from 4.529 to 4.277 and 3.529 to 3.277, respectively. As shown in Table 3, the dielectric and optical dielectric constants are affected by Na^+^ ion concentration and strongly depend on the value of the refractive index. However, with increasing amounts of sodium oxide, the metallization values of the current glass samples were found to be less than unity, confirming their nonmetallic characteristics and proving that these samples can be used as nonlinear optical materials [39].

Several parameters, including the optical band gap (E_opt_) and Urbach energy (ΔE), were considered essential optical parameters to characterize the glass material. The Davis–Mott relation was used to calculate the optical energy gaps for all samples [40,41].
(17)$$\alpha \left(hv\right)={\left(B\;(hv-E_{\mathrm{opt}})\right)}^{n}/hv$$
where *α*, *B*, E_g_, h and *v* are the absorption coefficient, a constant depending on the glass composition, the optical energy band gap, h is the Planck constant, and ν is the photon’s frequency, respectively. n = 2 for the indirect transition mechanism of electrons.

The optical energy gap (E_opt_) for the glass samples was obtained by extrapolating the linear region of (*α*(hν)hν)^1/2^ vs. (hν) to (*α*(hν)hν)^1/2^ = 0, as shown in Figure 2. As the Na_2_O content was increased, the energy gap was shown to reduce from 3.006 to 2.659 eV. This effect could be explained by an increase in the nonbridging oxygen in the glass, which causes it to weaken [42].

The Urbach energy (ΔE), or the width of localized states, is used to calculate the atomic structure’s disorder degree, which is represented by the following formula [43,44]:(18)$$\alpha \left(hv\right)=\beta \;\exp\left(\frac{hv}{\Delta E}\right)$$
where (β) is constant.

The reciprocal of the linear part’s slopes from the plot of ln(*α*) against (h*v*), as shown in Figure 3, was used to estimate these values. Table 2 shows the values of ΔE. As shown in Table 2, the values of ΔE were 0.3726 and 0.4804 eV for the samples TPN1 and TPN3, respectively. The sample TPN2 had the lowest value of ΔE among all samples. At a Na_2_O concentration of 20 mol%, this leads to a distortion in the TeO_4_ tbp units [38] and a decrease the atomic structure’s disorder degree. Otherwise, the sample TPN3 had the highest ΔE, referring to increases in the structure’s disorder degree.

### 3.2. Radiation Shielding Properties

Figure 4a shows the variation in mass attenuation coefficient (MAC) values for the glass samples investigated. Table 4 shows the MAC estimated using Phy-x and MIKE software for the samples under investigation. Linear attenuation coefficients (LACs) were estimated using Equation (11). As demonstrated in Table 4, there was good agreement between the calculated values. Figure 4 depicts the MAC behavior of the TPN glass system at energies ranging from 0.015 MeV to 15 MeV. As seen in Figure 4, the attenuation factor, MAC, had the highest values at lower energies and rapidly decreased as the photon energy increased. The TPN1, TPN2, and TPN3 glasses had MAC values of 33.383, 32.279, and 31.094 cm^2^/g, respectively, at an energy of 15 keV. The highest recorded values at lower energies were mainly due to the photoelectric interaction process. The MAC was highly dependent on the atomic number (i.e., Z^α^, where α = 4–5). This indicates the rationale for decreasing the attenuation coefficient as the Te concentration decreased. At lower energies, the effect of the K-absorption edge can be seen at a photon energy of 40 keV, which caused the discontinuity of the attenuation curve, which had a great effect on the attenuation efficiency in the lower energy range. The graphic illustration indicates that replacing TeO_2_ with Na_2_O generated a decrease in the attenuation factor. The calculated linear attenuation coefficients (LACs) for the TPN glass system using the MAC and the measured density are shown in Figure 4b. The maximum recorded LAC values for the TPN, TPN2, and TPN3 glasses were 153.63, 140.22, and 129.01 cm^−1^, respectively. Along with the reduction in the MAC and LAC due to the replacement of TeO_2_, as illustrated in Table 4, which can result in decreases in the shielding efficiency, the enhancement of optical properties and thermal stability is also of equal importance in obtaining optimum performance for shielding materials. It is therefore possible to enhance thermal stability and optical properties while maintaining acceptable shielding performance by selecting the optimum concentration of the modifier.

Figure 5a–c show the HVL, TVL, and MFP values of the prepared glasses. At low energy photons, these values were extremely small, and they became even smaller as the concentration of TeO_2_ increased. The HVL and TVL values for all glass samples were nearly constant up to 0.1 MeV before rapidly increasing and reaching a peak value of 7 MeV. The variations in these values for the existing glass samples can be expressed in terms of photon interactions in this energy range. The denser the sample was, the lower the values of MFP, HVL, and TVL. As shown in Figure 5, the photoelectric absorption had a great effect on these radiation shielding parameters in the lower energy range. The Compton and pair production processes were responsible for the plateau and the decrease in these values at higher energies. As shown in Figure 5, the sample TPN1 showed the lowest HVL, TVL, and MFP values and the highest densities and mass attenuation coefficients among the samples. Furthermore, the prepared glass system was compared with some common standard materials available commercially and widely used in medical applications [26]. TPN1 showed better performance than the commercial glasses RS253-G18 and RS360, while its performance was worse than that of RS520, as shown in Figure 6a,b. This was obviously due to the high content of lead oxide (71%), which made it more efficient than the others. Considering the toxicity of lead oxide, the prepared glasses have the superior ability to be used as an alternative shielding material in medical applications such as shielding glass windows as well as shielding materials directly used on patients undergoing X-ray procedures for diagnostic and interventional purposes. These findings are consistent with other findings in the literature [6,7].

The calculated values of the effective atomic number (Z_eff_) and the effective electron number (N_eff_) of TPN glass systems are shown in Figure 7a,b and Table 5. These values show that Z_eff_ and N_eff_ both had a significant dependence on photon energy and the glass material composition. The recorded values of Z_eff_ were in the range of 17 to 50, while those of N_eff_ ranged from 2.7 to 8.11 × 10^23^ electrons per gram. The discontinuity that appeared in both curves was mainly due to the influence of the K-shell absorption of tellurium (atomic number 52) in the energy range between 0.02 and 0.2 MeV. At these energies, both parameters reached their maximum values. There was a clear dependence on the elemental atomic number and photon energy in the region where the photoelectric interaction was the dominant effect. The lowest values were recorded in this region. The gradual increase in Z_eff_ and N_eff_ was due to the effect of Compton and pair formation interactions. The TPN1 glass had the highest Z_eff_ values, while TPN3 showed the lowest values. Because N_eff_ is directly proportional to the effective atomic number and inversely proportional to the mean atomic mass of the proposed shielding material, TPN3 had the highest N_eff_ values compared to the other samples.

The shielding effectiveness of the prepared glasses can also be investigated by using a term called radiation protection efficiency (RPE) [21].
(19)$$\mathrm{RPE}\%=1-e^{-\mu \rho }$$

Figure 8a shows the RPE percentages of the prepared glasses with a thickness of 1 cm at gamma-ray energies ranging from 0.015 to 0.2 MeV. With increasing photon energy, the RPE percentages decreased from 100 to 56.9% for TPN1, 100 to 54.06% for TPN2, and 100 to 51.63% for TPN3. As shown in Figure 8b, increasing Na_2_O concentrations and increasing oxygen molar volumes led to a decrease in RPE. As shown in Figure 8a, 1 cm of prepared glass had high shielding efficiency for energies up to 100 keV. For higher-energy applications, the effective thickness must be increased.

## 4. Conclusions

Increases in Na_2_O concentration resulted in a decrease in the density, optical energy gap, and refractive index of the TPN glass system, while the thermal stability was increased. There was a significant increase in nonbridging oxygen with the TeO_3_ phase and effective electron number (N_eff_) as a result of Na_2_O incorporation into TeO_2_/P_2_O_5_. The HVL, TVL, and MFP values decreased as the TeO_2_ increased. The energy gap, oxygen packing density (OPD), and linear refractive index (n) values decreased as the Na^+^ ion concentrations increased due to the increase in the concentration of nonbridging oxygen. On the other hand, the Urbach energy (E_u_), molar volume (V_M_), and oxygen molar volume (V_O_) increased as the concentration of sodium oxide increased. The metallization (M) values indicated the non-crystallized nature of the prepared glasses. TPN3 had the greatest absorbance values in the visible spectrum of light, while TPN2 had the lowest absorbance and lowest Urbach energy with the highest molar refractivity and electronic polarizability, implying that it is suitable for optical applications. When comparing the shielding performance of the prepared glasses to commonly used standard shielding materials, the results were satisfactory. With a high concentration of TeO_2_ and the right amount of Na_2_O in large bulk glasses, we were able to maintain the glass’s promising shielding effectiveness while also maintaining good thermal stability and good optical properties, which makes it a good choice for both shielding and optical use.

## Figures and Tables

**Figure 1 materials-15-03172-f001:**
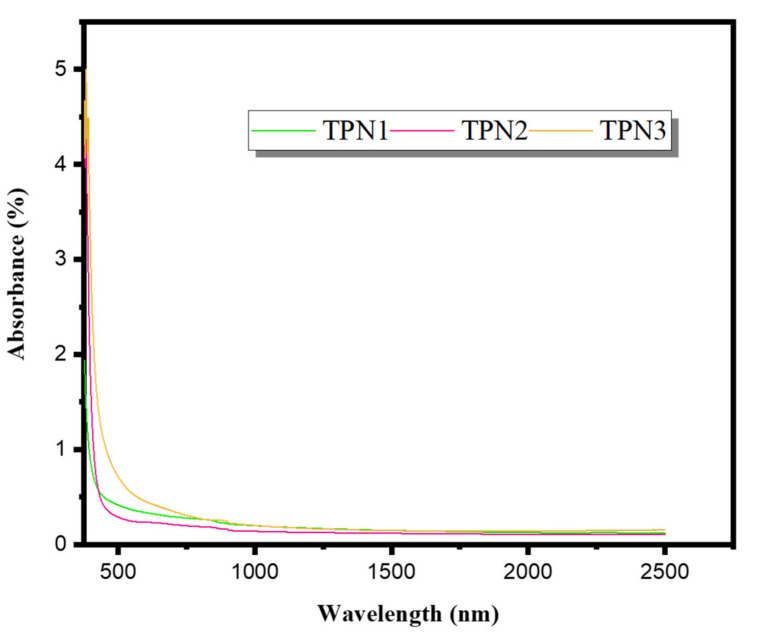
Absorbance spectroscopy for different compositions of TPN.

**Figure 2 materials-15-03172-f002:**
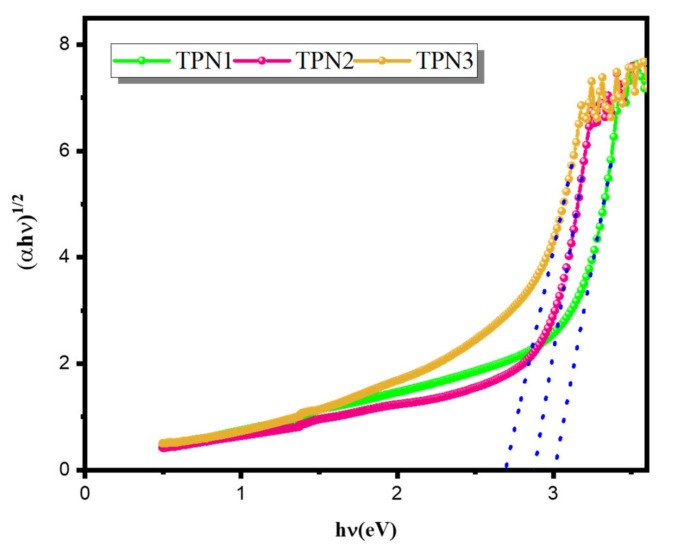
Plot of (αh*v*)^1/2^ as a function of the photon energy (h*v*) of prepared glasses.

**Figure 3 materials-15-03172-f003:**
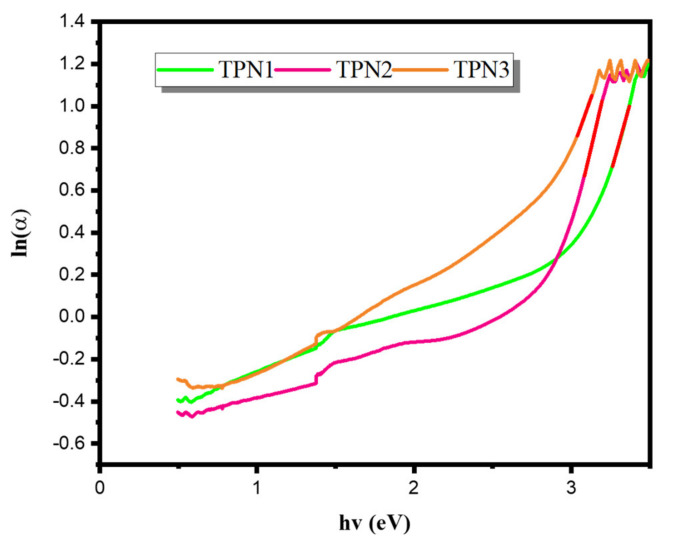
Plot of ln(α) as a function of the photon energy (h*v*) of prepared glasses.

**Figure 4 materials-15-03172-f004:**
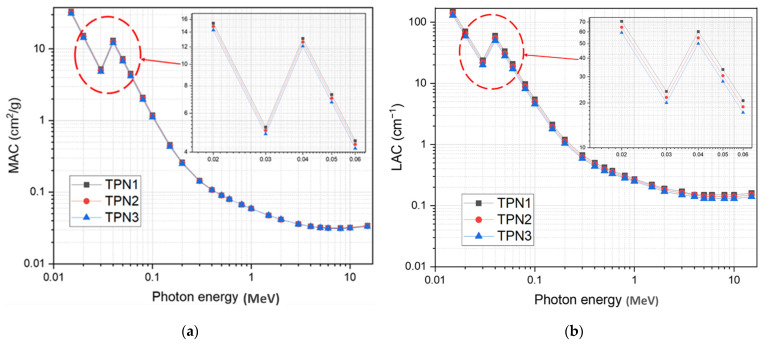
(**a**) The mass attenuation coefficient of prepared glasses; (**b**) The linear attenuation coefficient of prepared glasses.

**Figure 5 materials-15-03172-f005:**
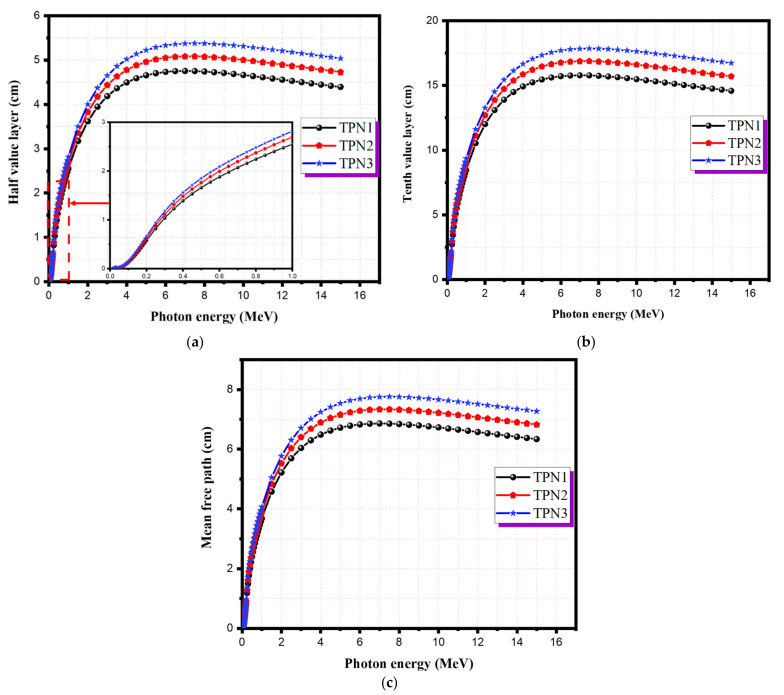
The shielding parameters of TPN systems: (**a**) HVL; (**b**) TVL; (**c**) MFP.

**Figure 6 materials-15-03172-f006:**
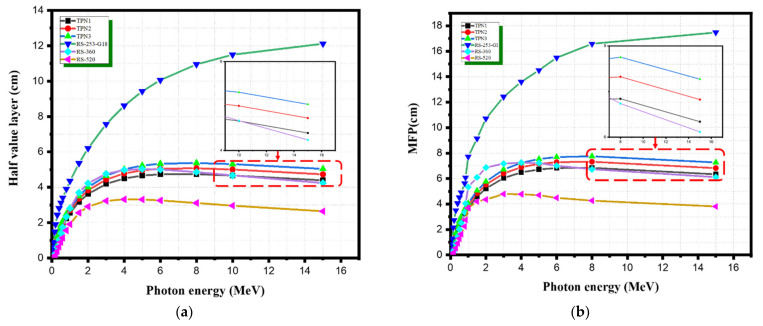
The shielding parameters for TPN and standard materials: (**a**) HVL; (**b**) MFP.

**Figure 7 materials-15-03172-f007:**
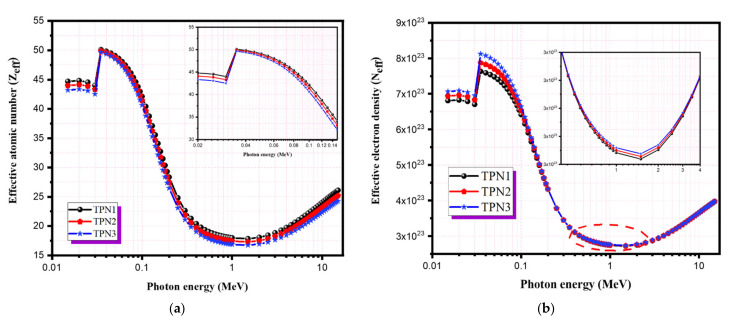
The radiation shielding parameters: (**a**) Effective atomic number (Z_eff_); (**b**) Effective electron number (N_eff_).

**Figure 8 materials-15-03172-f008:**
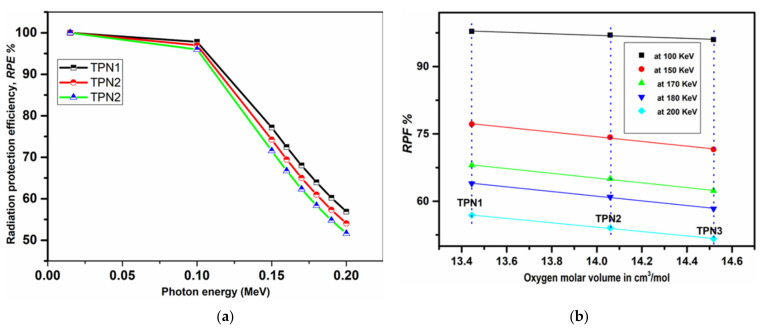
(**a**) RPE% with photon energy (in MeV) of fabricated glasses; (**b**) RPE% with oxygen molar volume of fabricated glasses.

**Table 1 materials-15-03172-t001:** The composition, density (ρ), and refractive index (n) of TPN glass system.

Sample Code	Composition (mol%)	Density in g cm^−3^ ± 0.037 [25]	Refractive Index ± 0.0002
TPN1	70TeO_2_–15P_2_O_5_–15Na_2_O	4.602	2.1281
TPN2	65TeO_2_–15P_2_O_5_–20Na_2_O	4.344	2.089
TPN3	60TeO_2_–15P_2_O_5_–25Na_2_O	4.149	2.0681

**Table 2 materials-15-03172-t002:** The molar volume (V_M_), oxygen molar volume (V_O_), oxygen packing density (OPD), energy gap (E_opt_), and Urbach energy (ΔE) of the fabricated glasses.

Sample Code	V_M_ (cm^3^·mol^−1^)	V_O_ (cm^3^·mol^−1^)	OPD (mol·dm^−3^)	Energy Gap, Eopt (eV) ±0.0047 eV	Urbach Energy, ΔE (eV) ±0.0016
TPN1	30.923	13.445	74.378	3.006	0.3726
TPN2	31.636	14.061	71.121	2.876	0.3146
TPN3	31.950	14.521	68.864	2.695	0.4804

**Table 3 materials-15-03172-t003:** The molar refractivity (R_m_), reflection loss (R_L_), electronic polarizability (α_m_), metallization (M), dielectric constant (ε), and optical dielectric constant (ε**_O_**) values of the studied glasses.

Sample Code	$R_{m}$ $\;(cm^{3}/\mathrm{mol})$	$R_{l}$ $\;(cm^{3}/\mathrm{mol})$	$\alpha_{m}$ $\;({Å}^{3})$	(M) (±0.001)	ε	ε_O_
TPN1	16.714	0.130	6.633	0.459	4.529	3.529
TPN2	16.722	0.124	6.635	0.471	4.364	3.364
TPN3	16.678	0.121	6.618	0.478	4.277	3.277

**Table 4 materials-15-03172-t004:** The calculated mass attenuation coefficient (MAC) and linear attenuation coefficient (LAC) values for TPN glass systems.

Photon Energy	MAC (cm^2^/g)						LAC (cm^−1^)					
	MIKE			Phy-X			MIKE			Phy-X		
	TPN1	TPN2	TPN3	TPN1	TPN2	TPN3	TPN1	TPN2	TPN3	TPN1	TPN2	TPN3
0.015	33.384	32.28	31.094	33.381	32.277	31.092	153.62	140.21	129	153.63	140.22	129.01
0.02	15.366	14.855	14.305	15.367	14.855	14.306	70.719	64.532	59.356	70.72	64.53	59.35
0.03	5.1889	5.0173	4.833	5.189	5.0170	4.833	23.878	21.794	20.051	23.88	21.79	20.05
0.04	13.100	12.612	12.088	13.098	12.609	12.085	60.275	54.775	50.142	60.29	54.79	50.15
0.05	7.2875	7.0182	6.7291	7.287	7.0180	6.7290	33.536	30.486	27.918	33.54	30.49	27.92
0.06	4.5018	4.3376	4.1612	4.502	4.3380	4.1610	20.718	18.843	17.266	20.72	18.84	17.26
0.08	2.1103	2.0362	1.9566	2.110	2.0360	1.9570	9.7112	8.8448	8.1175	9.710	8.850	8.120
0.1	1.1904	1.1508	1.1083	1.190	1.1510	1.1080	5.4778	4.9988	4.5981	5.480	5.000	4.600
0.15	0.4586	0.4462	0.4329	0.459	0.4460	0.4330	2.1106	1.9383	1.796	2.110	1.940	1.800
0.2	0.2612	0.2558	0.2501	0.261	0.2560	0.2500	1.2019	1.1112	1.0375	1.200	1.110	1.040
0.3	0.1449	0.1434	0.1417	0.145	0.1430	0.1420	0.6668	0.6227	0.5879	0.670	0.620	0.590
0.4	0.1083	0.1077	0.1071	0.108	0.1080	0.1070	0.4985	0.468	0.4445	0.500	0.470	0.440
0.5	0.0907	0.0905	0.0903	0.091	0.0910	0.0900	0.4176	0.3932	0.3746	0.420	0.390	0.370
0.6	0.0801	0.0800	0.0800	0.080	0.0800	0.0800	0.3686	0.3477	0.3318	0.370	0.350	0.330
0.8	0.0671	0.0672	0.0673	0.067	0.0670	0.0670	0.309	0.292	0.2792	0.310	0.290	0.280
1	0.0591	0.0592	0.0593	0.059	0.0590	0.0590	0.2718	0.257	0.246	0.270	0.260	0.250
1.5	0.0475	0.0476	0.0477	0.047	0.0480	0.0480	0.2184	0.2067	0.1979	0.220	0.210	0.200
2	0.0416	0.0417	0.0418	0.042	0.0420	0.0420	0.1915	0.1811	0.1733	0.190	0.180	0.170
3	0.0360	0.0359	0.0359	0.036	0.0360	0.0360	0.1655	0.1561	0.149	0.170	0.160	0.150
4	0.0335	0.0334	0.0333	0.033	0.0330	0.0330	0.1541	0.145	0.1381	0.150	0.150	0.140
5	0.0323	0.0322	0.0320	0.032	0.0320	0.0320	0.1488	0.1397	0.1327	0.150	0.140	0.130
6	0.0318	0.0316	0.0313	0.032	0.0320	0.0310	0.1464	0.1372	0.13	0.150	0.140	0.130
8	0.0318	0.0314	0.0311	0.032	0.0310	0.0310	0.1461	0.1365	0.129	0.150	0.140	0.130
10	0.0323	0.0319	0.0315	0.032	0.0320	0.0310	0.1486	0.1385	0.1305	0.150	0.140	0.130
15	0.0343	0.0337	0.0331	0.034	0.0340	0.0330	0.1577	0.1465	0.1375	0.160	0.150	0.140

**Table 5 materials-15-03172-t005:** Effective atomic number and effective electron density values for TNP glasses.

Photon Energy	Z_eff_						N_eff_ ×10^23^ (Electrons. Gram^−1^)					
	MIKE			Phy-X			MIKE			Phy-X		
	TPN1	TPN2	TPN3	TPN1	TPN2	TPN3	TPN1	TPN2	TPN3	TPN1	TPN2	TPN3
0.015	44.697	43.985	43.191	44.7	43.98	43.19	6.81	6.94	7.07	6.80	6.94	7.10
0.02	44.813	44.121	43.35	44.81	44.12	43.35	6.83	6.96	7.09	6.80	6.96	7.10
0.03	44.015	43.304	42.512	44.02	43.3	42.51	6.71	6.83	6.95	6.70	6.83	7.00
0.04	49.833	49.614	49.363	49.83	49.61	49.36	7.59	7.83	8.07	7.60	7.83	8.10
0.05	49.082	48.806	48.489	49.08	48.81	48.49	7.48	7.69	7.93	7.50	7.70	7.90
0.06	48.066	47.714	47.313	48.07	47.71	47.31	7.32	7.53	7.74	7.30	7.53	7.70
0.08	45.365	44.834	44.235	45.37	44.83	44.24	6.91	7.07	7.24	6.90	7.07	7.20
0.1	42.119	41.416	40.634	42.12	41.41	40.63	6.42	6.53	6.65	6.40	6.53	6.60
0.15	34.100	33.172	32.176	34.10	33.17	32.18	5.19	5.23	5.26	5.20	5.23	5.30
0.2	28.360	27.44	26.478	28.36	27.44	26.48	4.32	4.33	4.33	4.30	4.33	4.30
0.3	22.653	21.877	21.085	22.65	21.88	21.09	3.45	3.45	3.45	3.50	3.45	3.40
0.4	20.403	19.72	19.028	20.40	19.72	19.03	3.11	3.11	3.11	3.10	3.11	3.10
0.5	19.359	18.726	18.087	19.36	18.73	18.09	2.95	2.95	2.96	2.90	2.95	3.00
0.6	18.804	18.199	17.591	18.80	18.20	17.59	2.87	2.87	2.88	2.90	2.87	2.90
0.8	18.253	17.677	17.099	18.25	17.68	17.10	2.78	2.79	2.79	2.80	2.79	2.80
1	17.994	17.432	16.869	17.99	17.43	16.87	2.74	2.75	2.76	2.80	2.77	2.80
1.5	17.846	17.293	16.738	17.85	17.29	16.74	2.72	2.73	2.74	2.70	2.75	2.80
2	18.064	17.498	16.931	18.06	17.50	16.93	2.75	2.76	2.77	2.80	2.76	2.80
3	18.853	18.244	17.632	18.85	18.24	17.63	2.87	2.88	2.88	2.90	2.88	2.90
4	19.750	19.095	18.435	19.75	19.10	18.44	3.01	3.01	3.02	3.00	3.01	3.00
5	20.632	19.934	19.228	20.63	19.93	19.23	3.14	3.15	3.15	3.10	3.14	3.10
6	21.442	20.707	19.961	21.44	20.70	19.96	3.27	3.27	3.27	3.30	3.27	3.30
8	22.86	22.065	21.255	22.86	22.06	21.25	3.48	3.48	3.48	3.50	3.48	3.50
10	24.024	23.186	22.327	24.02	23.19	22.33	3.66	3.66	3.65	3.70	3.66	3.70
15	26.105	25.200	24.267	26.11	25.20	24.27	3.98	3.98	3.97	4.00	3.98	4.00

## Data Availability

Not applicable.

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
