# Peer review of "Evaluation of the Radiation Shielding Properties of a Tellurite Glass System Modified with Sodium Oxide"

_materials, 2022, doi:10.3390/ma15093172_

Round 1
Reviewer 1 Report
The paper by Hussein et al gives interesting results concerning radiation shielding properties of tellurite based glasses containing different amounts of sodium. In my opinion this work deserves to be published. However several corrections are required taking into account the following items:
- Section 2.1 line 113: ref 46 must be renumbered since it appears between refs 29 and 30.
- Section 3.1 line 178-179: given values do not match with Table 2 values.
- Lines 187: "increases" instead of "decreases". TPN3 sample shows the greatest absorbance.
- Section 3.1 lines 192-206: this part must be carefully reviewed. Given values in the text do not agree at all with Table 3 values (which are correct). The same for line 218 (and Table 2 values).
- Table 3: There is no clear trend for molar polarizability values. Please explain.
- Line 234: How are calculated LAC values?
- Line 246: 15 MeV instead of 10.
- Line 248: "decrease" instead of "increase".
- Line 251: unit for LAC is wrong.
- Table 4 and 5 (instead of 6): Please give the units for energy, MAC, LAC, Zeff and Neff.
- Fig. 4: MeV instead of keV.
- Table 5 caption: what is PCKNT?
- Conclusion lines 319-320: assertions are wrong. Please correct.
Reviewer 2 Report
In this manuscript, the authors presented a manuscript titled as “Evaluation of radiation shielding properties of tellurite glass 2 system modified with sodium oxide” and studied on the optical and the radiation shielding properties of tellurite glasses with a glass composition of (85-x) TeO2-10P2O5-xNa2O mol % (where x = 15, 20, 25). The authors discussed the optical properties and radiation shielding capabilities of the glass series using PHY-X/ PSD and MIKE software whose structural properties were studied before. My first impression is that the authors made a successful work. I mentioned that introduction part is well prepared. The data obtained from the study were compared with the literature. Radiation properties of ternary glasses were considered theoretically and their calculations were carried out at different energy ranges in this study. The work is written in an understandable way. The scientific content of the paper looks good. I think the manuscript can be accepted, provided that the authors make some corrections. I suggest the authors make the following corrections:
1- The terms “Bioactive glass” should be removed from keywords. Because there are no information or related topics about on the whole study.
2- It may be better visually if the given equations in “Materials in Methods” section are written in the same format and size.
3- In Results and Discussion section the authors used this sentence:
“The measured density data for the fabricated glasses was reported by Yousef [25].”
Since the authors have already made this statement before in Materials and Methods section, this sentence is not needed. The reference [25] can be given to the density section in Table 1 instead.
4- It can be good if the error calculation is made in the calculation of the refractive index like density values.
5-The authors declared that they used the range of 190-2500 nm for optical properties. However, Figure 1 is the absorbance spectrum in the 250-2500 nm range. Therefore, they need to correct the range where they take optical measurements. Also, the thickness of the lines in the figure makes it difficult to read the absorbance values of the samples. It can be more understandable if thinner lines are preferred and I think there is no need for the dashed lines in the background of Figure 1.
6- “The optical energy gap (Eopt) for glass samples was obtained by extrapolating the linear region of (kv)1/2 vs (hv) to (kv)1/2 = 0, as shown in Figure 2.” What is k?
7- Is it possible for the authors to explain the chaotic variation in Urbach energy?
8- The unit of Photon energy given in Figure 4 (a) and (b) should be checked. Is it MeV or keV?
9- Some typos should be corrected.
10- References should be a same format.
Reviewer 3 Report
The work seams new and interesting but major revisions are necessary, as follows.
- Keywords: I suggest you to replace the keyword “Bioactive glass” which is inappropriate. I suggest “tellurite glass” or “phospho-tellurite glass” instead.
- Introduction: “Phosphate-based glasses have aroused the interest of researchers due to their distinctive properties such as high thermal expansion coefficients” – High thermal expansion coefficients comparatively to? - As I know the thermal expansion coefficient of usual phosphate glass is similar to usual silicate glass, but phosphate glass possess other high properties, such as optical properties or high doping possibilities. Please improve the text.
- Please describe the method of obtaining the glass, including temperatures and time of thermal treatment. Also include the furnace and the crucibles used.
- It is not clear what is the novelty of your work. Please insert some sentences in the end of Introduction chapter which clearly describe the novelty of your work. The glass are of a new composition, for example, or is this the first time when MIKE is used in this purpose?
- Please insert where available the apparatus error, for example for the refractive index measurements.
- English translation problems: Materials and Methods chapter: “The shielding parameters were compared with compared with commonly used commerical standard radiation shielding materials”: “compared with compared”? “commerical”? Other problem in 2.1. Optical properties chapter: “Where Mw is defined as the total molar weight of sample composition sample”: “sample composition sample”? Please check again the English translation for all manuscript.
- Please insert References when introducing equations, when they are not original, such as equation (1). Check the all text.
- What is the percent RL? (equation (5)) Are you sure, because the equation is not percentual? Explain in text the annotation RL. Please do not introduce in text any abbreviation without explaining it. Another example: M from equation (6). The abbreviation/annotation of dielectric constant is not usual. Please change it to the usual ε. OPD abbreviation of oxygen packing density is also introduced in text without explaining. Please check the entire text and do the necessary improvements.
- 2.1.Chapter: “Where I0, Ix, μ, and x denote the intensity of incident radiation, attenuated radiation intensity, linear attenuation coefficient, and attenuated material thickness, respectively.
When the photon beam is monochromatic, this rule can be applies to all materials. A ratio between and the density of the shielding material determines the mass attenuation coefficient (MAC) in cm2 .g(1)”. In this paragraph multiple errors exist: i) “attenuated material” is in fact attenuating material; ii) I doubt this rule can be applied to all materials, so, please erase this part of the sentence; iii) A ration between what?; iv) measurement units are not properly written. Please improve the paragraph and check the all text for such errors.
- 3.1. Chapter: “The refractive index decreases from 2.128 to 2.068, while the density decreases from 4.602 to 4.149 gm.cm-3. This is due to the presence of TeO2, which is heavier than Na2O”: I suggest you to change these sentences, because the increase of refractive index and of density is due to TeO2 increasing, and the decrease of them are due to Na2O increasing which is lighter.
- Tables 1 and 2. I suggest to introduce uncertainties for all values, including the refractive index and bandgap energy, since they cannot be precisely measured or calculate from graphs. Attention to measurement units in Table 2, correct them.
- Page 5 of PDF file: “Both VM and VO are proportional to the spatial distributions of oxygen in the glass matrix, increasing from 30.925 to 31.94 cm3 and 13.445 to 14.518 cm3.mol-1 correspondingly. When the Na2O concentration is increased from 15 to 25 mol%, the OPD value has dropped from 74.37 to 68.86 gm.atmL-1”: The values in this paragraph are different from those in Table 2. This is an error that cannot be permitted! Please be more careful!! What is gm.atmL-1 and why you inserted a different measurement unit in Table 2?
- Page 5 of PDF file: “Furthermore, as illustrated in Figure 1, increasing the concentration of sodium oxide (Na2O), decreases the absorbance”: Fig. 1 shows otherwise, since TPN3 has the greater absorbance. Please change in text.
“TPN3 has the greatest absorbance values in the visible spectrum of light, implying that it is suitable for optical applications“: Why? Seams to me that the lowest absorbance indicates the most suitable material for optical applications. Please improve the text.
- Page 6 of PDF file: “The refractive index decreased from 2.1281 to 2.0681 as the concentration of Na2O increased, whereas the dielectric constant increased from 6.031 to 6.443. The optical dielectric constant and refraction loss inceased with the increases of Na2O from 3.0401 to 3.1616 and 0.112 to 0.117, respectively. The density of glass influences its refractive index. With increasing density, the refractive index increases [39]. The dielectric and optical dielectric constants are affected by ion concentration. These values increase as the ion concentrations increase and with the sodium concentration increases [40]. However, with increasing amounts of sodium oxide, the molar refractivity and molar polarizability of the glass in this study increased from 15.564 to 16.392 and 6.176 to 6.504, respectively”: This paragraph too has multiple problems: i) again the values in text are not the same with the values in table 3! Did anyone of authors check this text for errors before submitting it to this Journal? ii) what do you mean by ion concentration”? Which ion? iii) where have you inserted the molar polarizability values, since in Table 3 there is no such property.
- Table 3 caption is inaccurate, since the font is not the same, the molar reflection is not the correct name, probably molar refractivity and some words are wrong capitalized.
- “The optical energy gap (Eopt) for glass samples was obtained by extrapolating the linear region of (kv)1/2 vs (hv) to (kv)1/2 = 0, as shown in Figure 2. With increasing the Na2O content, the energy gap was shown to reduce from 3.403 to 3.279 eV”: Again multiple errors: i) “region of (kv)1/2 vs (hv) to (kv)1/2 = 0” is inaccurate, since you plotted αhv against hv. ii) The values are again different from those in Table 2!!! iii) Why did you choose n = 2 in the case of glasses?
- Regarding Chapter 3.1. I don’t see the necessity to keep Tables 4 and 6 (where is Table 5?) since you have inserted in Figs. 4-7/7bis these data. You have two Fig. 7 in the paper. Another prove that nobody checked the manuscript before sending it to be published!
- You must change the Conclusions chapter accordingly to above observations.
Round 2
Reviewer 1 Report
The paper by Hussein et al can be accepted for publication taking into account a minor correction:
-Conclusion line 353: "decreased" instead of "increased" (see optical band gap values in Table 2).
